# Usefulness of Carnitine Supplementation for the Complications of Liver Cirrhosis

**DOI:** 10.3390/nu12071915

**Published:** 2020-06-29

**Authors:** Tatsunori Hanai, Makoto Shiraki, Kenji Imai, Atsushi Suetugu, Koji Takai, Masahito Shimizu

**Affiliations:** Department of Gastroenterology/Internal Medicine, Gifu University Graduate School of Medicine, 1-1 Yanagido, Gifu 501-1194, Japan; mshiraki-gif@umin.ac.jp (M.S.); ikenji@gifu-u.ac.jp (K.I.); asue@gifu-u.ac.jp (A.S.); koz@gifu-u.ac.jp (K.T.); shimim-gif@umin.ac.jp (M.S.)

**Keywords:** carnitine, hepatic encephalopathy, liver cirrhosis, muscle cramps, sarcopenia

## Abstract

Carnitine is a vitamin-like substance that regulates lipid metabolism and energy production. Carnitine homeostasis is mainly regulated by dietary intake and biosynthesis in the organs, including the skeletal muscle and the liver. Therefore, liver cirrhotic patients with reduced food intake, malnutrition, biosynthetic disorder, and poor storage capacity of carnitine in the skeletal muscle and liver are more likely to experience carnitine deficiency. In particular, liver cirrhotic patients with sarcopenia are at a high risk for developing carnitine deficiency. Carnitine deficiency impairs the important metabolic processes of the liver, such as gluconeogenesis, fatty acid metabolism, albumin biosynthesis, and ammonia detoxification by the urea cycle, and causes hypoalbuminemia and hyperammonemia. Carnitine deficiency should be suspected in liver cirrhotic patients with severe malaise, hepatic encephalopathy, sarcopenia, muscle cramps, and so on. Importantly, the blood carnitine level does not always decrease in patients with liver cirrhosis, and it sometimes exceeds the normal level. Therefore, patients with liver cirrhosis should be treated as if they are in a state of relative carnitine deficiency at the liver, skeletal muscle, and mitochondrial levels, even if the blood carnitine level is not decreased. Recent clinical trials have revealed the effectiveness of carnitine supplementation for the complications of liver cirrhosis, such as hepatic encephalopathy, sarcopenia, and muscle cramps. In conclusion, carnitine deficiency is not always rare in liver cirrhosis, and it requires constant attention in the daily medical care of this disease. Carnitine supplementation might be an important strategy for improving the quality of life of patients with liver cirrhosis.

## 1. Introduction

Carnitine is a vitamin-like biofactor involved in fatty acid and energy metabolism. The dysfunction of the liver causes several metabolic disorders because hepatocytes play a central role in glucose, lipid, and protein metabolism. Carnitine deficiency is frequently observed in patients with liver cirrhosis (LC), especially in those with sarcopenia and malnutrition. Recent clinical evidence has revealed the possibility that supplementation with carnitine is effective in improving the complications of LC, including hepatic encephalopathy, sarcopenia, and muscle cramps. In this review, we summarized the physiological function and homeostasis of carnitine. The effectiveness of carnitine in improving the complications of LC was also investigated, based on the recent clinical trials.

## 2. Physiological Function of Carnitine

Carnitine (β-hydroxy-γ-*N*-trimethylaminobutyric acid), which was discovered in bovine muscle extract, is a vitamin-like substance or micronutrient that acts as a necessary ingredient for body functioning and metabolism. Carnitine can be obtained from animal sources, such as dairy products, meat, poultry, and fish, in daily meals. In humans, endogenous carnitine is synthesized in the liver, kidney, and brain from two essential amino acids (methionine and lysine), three vitamins (ascorbic acid, niacin, and vitamin B6), and iron. Carnitine consists of L-bodies and D-bodies, with only the L-bodies (L-carnitine/levocarnitine) showing bioactivity; thus, only the L-bodies can be clinically applied in medical treatment [1,2].

The critical role of carnitine in fatty acid metabolism is well reported. Carnitine binds to long-chain acyl-coenzyme A (CoA) and converts it to acylcarnitine, which is transported to the mitochondria. Carnitine is, thus, essential for transferring long-chain fatty acids to the mitochondrial matrix and subsequently promotes energy metabolism and adenosine triphosphate production by activating the β-oxidation of fatty acids [3]. Carnitine is also required for the outflow of acyl-CoA from the mitochondria. Therefore, in addition to fatty acid metabolism, carnitine exerts a good influence on gluconeogenesis, the urea cycle, the glycolysis system, and the tricarboxylic acid cycle, through the regulation of acyl-CoA and pool of free CoA levels in the mitochondria [4]. Moreover, carnitine improves the inflammation, oxidative stress, and function of the biomembrane [5,6,7]. The physiological functions of carnitine are listed in Table 1.

## 3. Homeostasis of Carnitine and Liver Disease

Generally, 75% of the daily carnitine requirement for a healthy life is supplied from the diet, and the remaining 25% is produced by the body, including the liver [6]. Carnitine is mainly present in the skeletal muscle, heart, and liver. Approximately 98% of the body’s carnitine content is stored in the muscular tissue of the skeletal muscle and myocardium, and only 0.6% exists in the blood [8,9]. Therefore, patients with sarcopenia, which reduces the skeletal muscle mass, easily develop carnitine deficiency because of poor retention. In addition, the serum carnitine level is not reflective of the tissue carnitine level, because only a low level of carnitine exists in the blood (Table 2) [8,10].

The carnitine level in the body is mainly determined by intake from food, the amount of biosynthesis, and its reabsorption at the renal tubule. Carnitine is abundantly present in lean meat and dairy products. Therefore, patients with protein restrictions, such as LC patients, may develop carnitine deficiency. In the body, carnitine is biosynthesized in the liver, brain, and kidney. Among these organs, the liver plays an important role in the production of carnitine; thus, LC patients with a decreased liver function reserve are more likely to experience carnitine deficiency [11]. The kidney is also an important organ that regulates the carnitine level, because 90% of the body’s carnitine content excreted from the glomerulus is reabsorbed at the renal tubule [12].

## 4. Diagnosis of Carnitine Deficiency

Carnitine deficiency is considered as a condition in which the concentrations of carnitine in both plasma and tissue cannot reach the level that exhibits normal functioning of the living body. Primary and secondary carnitine deficiencies have been reported, and LC belongs to the latter [13]. Primary carnitine deficiency is associated with hepatic steatosis, hyperammonemia, and skeletal myopathy, which shows a close relationship between carnitine deficiency and dysfunction of the liver and skeletal muscle [14]. Generally, patients with carnitine deficiency show nonspecific symptoms, including general malaise, fatigue, impaired consciousness, encephalopathy, and cramps, which are also observed in patients with LC (Table 3). When carnitine deficiency associated with LC is suspected, clinical examinations on liver dysfunction, electrolyte abnormality, hyperammonemia, metabolic acidosis, and hypoglycemia should be performed. Little evidence exists regarding the frequency of carnitine deficiency in patients with LC. Our previous study, involving 70 cirrhosis patients, showed that the free carnitine concentration was in the normal range of 53.2 ±2.6 μmol/L [10], suggesting the difficulty of diagnosing carnitine deficiency based only on blood test results.

Carnitine exists as free carnitine and acylcarnitine fractions in blood samples. The tandem mass analysis and enzyme cycling method are used for the measurement of serum carnitine levels. The acylcarnitine analysis using tandem mass spectrometry is essential for the diagnosis of congenital metabolic disorders in the pediatric field. To diagnose carnitine deficiency, the blood levels of total carnitine and free carnitine were measured by using the enzyme cycling method, and this procedure has been covered by medical insurance in Japan since 2018.

## 5. Chronic Liver Disease and Carnitine Deficiency

Patients with LC, especially those with sarcopenia, are at high risk of developing carnitine deficiency. In 1977, Rudman et al. reported that carnitine depletion is common in patients with advanced LC [15]. On the other hand, several studies have shown that serum carnitine levels do not always decrease in patients with LC, and they are sometimes within the normal range or higher [10,11,16,17]. We have reported that serum carnitine concentrations (total carnitine, free carnitine, and acylcarnitine) were within the normal range in the majority of Japanese LC patients [10]. This might be explained by losses of skeletal muscle mass and hepatocyte associated with LC, which can cause an outflow of intracellular carnitine (carnitine in tissues) to pool into the blood. Therefore, in patients with LC or liver failure, it is difficult to diagnose carnitine deficiency based on the measurement of blood carnitine concentrations, and it is unavoidable to rely on the diagnosis based on clinical symptoms and general clinical laboratory findings.

Recently, Miyaaki et al. have reported an interesting finding that carnitine fraction levels were positively correlated with liver function reserve and physical symptoms of LC, although the blood level of free carnitine was within the normal range, measured by using tandem mass spectrometry [17]. This report may suggest that profiling of serum carnitine fraction is useful to determine potential carnitine deficiency in chronic liver disease patients.

## 6. Carnitine Supplementation and LC

If carnitine deficiency is diagnosed based on clinical symptoms or laboratory findings, carnitine replacement therapy should be administered to improve the clinical symptoms or laboratory abnormalities in LC patients. Carnitine supplementation is provided to address the malnutrition and metabolic disorders of patients, because the liver is the major organ that regulates nutrition and metabolism. Patients with LC frequently suffer from protein-energy malnutrition, which is closely associated with a high mortality [18,19]. Importantly, a recent clinical trial showed evidence that carnitine supplementation significantly improves energy metabolism and suppresses systemic inflammation in patients with LC [20]. Carnitine deficiency also impairs important metabolic processes of the liver, such as gluconeogenesis, fatty acid metabolism, albumin biosynthesis, and reduction of ammonia by the urea cycle. These metabolic abnormalities are involved in hypoalbuminemia and hyperammonemia, which induce several complications of LC, including hepatic encephalopathy, sarcopenia, and muscle cramps (Figure 1). The beneficial effects of carnitine supplementation on these LC complications are described below.

In addition, there is a need to consider the appropriate amount of carnitine supplementation and note the adverse effects that have been associated with high-dose carnitine supplementation. There is still controversy concerning the appropriate amount of carnitine intake. Research has suggested that high-dose carnitine supplementation (approximately 3 g/day) may cause several adverse effects including nausea, vomiting, abdominal cramps, diarrhea, muscle weakness, seizures, atherosclerosis, and cardiovascular disease [21,22,23,24,25]. Table 4 summarizes the information on carnitine supplementation dose, duration, and outcomes reported by previous studies.

## 7. Carnitine Supplementation and Hepatic Encephalopathy

Hyperammonemia is critically involved in the development of hepatic encephalopathy [38]. We have reported that L-carnitine supplementation significantly reduced serum ammonia levels in LC patients [10]. Nojiri et al. conducted a randomized trial and found that L-carnitine supplementation improved hyperammonemia and covert encephalopathy in patients with LC [33]. In this trial, the results of the neuropsychiatric test (number connection test B) improved after L-carnitine supplementation [33]. Importantly, a recent systematic review, which analyzed seven randomized controlled trials involving 660 participants, revealed that acetyl-L-carnitine was effective in improving serum ammonia levels and the connection test completion time [39].

The effects of carnitine supplementation on the improvement of hepatic encephalopathy were revealed by several randomized and controlled clinical trials conducted by Malaguarnera et al. [26,27,28,29,30,31,32]. In these reports, patients with various grades of hepatic encephalopathy (from minimal hepatic encephalopathy to coma) were enrolled, and supplementation with L-carnitine or acetyl-L-carnitine significantly improved the parameters associated with hepatic encephalopathy [26,27,28,29,30,31,32]. The results of these trials may show the usefulness of carnitine supplementation for the treatment of hepatic encephalopathy. However, large-scale multi-center studies will be necessary to verify this possibility, because the abovementioned clinical trials were conducted by only one research team and thus may cause a high risk of bias [40].

## 8. Carnitine Supplementation and Sarcopenia

Sarcopenia, the loss of skeletal muscle mass and strength, is common in patients with LC and is associated with the poor survival rate of these patients [19,41]. Therefore, there is a critical need to develop effective treatments for this condition. Multiple mechanisms including increased ammonia levels and malnutrition are involved in the development of sarcopenia in LC patients; thus, carnitine is considered a therapeutic agent to improve sarcopenia [42,43]. Malaguarnera et al. reported that the administration of L-carnitine (4000 mg/day) suppressed skeletal muscle loss in patients with LC [31]. Hiramatsu et al. also revealed that L-carnitine supplementation can suppress the progression of sarcopenia through the improvement of hyperammonemia in patients with LC [35]. In this study, the administration of a high dose of L-carnitine (≥1274 mg/day) was associated with increased skeletal muscle mass [35]. The results of this retrospective study might indicate that carnitine supplementation is useful for the therapeutic management of sarcopenia in patients with LC, at least, by targeting hyperammonemia.

On the other hand, Ohara et al. reported an interesting finding that L-carnitine supplementation had a preventive effect on skeletal muscle depletion, even in LC patients without decreased ammonia levels [34]. These findings might be explained by the anti-inflammatory effect and antioxidant activity of carnitine, because systemic inflammation with an increase in radical species is involved in sarcopenia development in patients with LC [42,44]. Carnitine is also considered to increase the skeletal muscle mass in patients with carnitine deficiencies and to alleviate muscle injury [45,46].

## 9. Carnitine Supplementation and Muscle Cramps

Muscle cramps are a major complication of LC that are related to a lower quality of life and sleep disturbance [47,48]. Decreased carnitine production due to the underutilization of methionine is considered to be involved in muscle cramps. Although the standard treatment for cirrhosis-related muscle cramps has not been established, carnitine supplementation is becoming widely used for controlling muscle cramps and improving the quality of life in these patients [48].

Nakanishi et al. performed a prospective study and found that a high dose of L-carnitine supplementation (1200 mg/day) significantly suppressed the incidence and the degree of muscle cramps in patients with LC [36]. Hiraoka et al. also reported that additional supplementation of L-carnitine (1000 mg/day) plus exercise (2000 steps/day) was effective in reducing muscle cramps in LC patients receiving branched chain amino acid preparation [37]. These reports may suggest that the usage of a high dose of carnitine and combining this drug with other treatments, such as nutrition therapy, exercise, and drug therapy, are useful strategies for reducing muscle cramps in LC patients.

## 10. Carnitine and Hepatocellular Carcinoma

Patients with LC are at a high risk for developing hepatocellular carcinoma (HCC) [49]. Carnitine might exert preventive effects on HCC development, because it has anti-inflammatory and antioxidant properties, which are both critical mechanisms of cancer prevention [50]. In a mouse model, the administration of L-carnitine prevented the progression of non-alcoholic steatohepatitis, and subsequent liver tumorigenesis, by suppressing oxidative stress and inflammation in the liver [51]. However, thus far, there has been no significant clinical evidence showing that carnitine supplementation can prevent liver carcinogenesis.

On the other hand, there are some important reports showing the relationship between alteration of acylcarnitine metabolism and steatosis-/obesity-related liver carcinogenesis [52,53,54]. Fujiwara et al. reported that metabolic reprogramming, including the accumulation of acylcarnitine, mediated by the downregulation of carnitine palmitoyltransferase-2, is involved in steatosis-/obesity-related hepatocarcinogenesis [52]. These findings may suggest that targeting the acylcarnitine metabolism is a potential strategy for the prevention and treatment of HCC, at least, caused by steatosis and obesity. At the same time, it is necessary to carefully observe the effect of carnitine supplementation itself on liver carcinogenesis in patients with LC.

## 11. Conclusions

Carnitine deficiency is not always rare in patients with LC, and it is always necessary to consider the condition, especially in patients with hepatic encephalopathy, sarcopenia, and muscle cramps. There is a possibility that patients with LC are in a state of relative carnitine deficiency at the liver, skeletal muscle, and mitochondrial levels, even if the blood carnitine concentration is in the normal range or higher. In patients with LC, it is difficult to diagnose carnitine deficiency by measuring its blood concentration; thus, it is necessary to make a comprehensive diagnosis based on clinical symptoms and general clinical laboratory findings. Carnitine deficiency may be diagnosed in some cases by the significant improvement in clinical symptoms and signs as a result of carnitine supplementation. Correctly diagnosing carnitine deficiency and applying appropriate interventions may lead to the improvement of the prognosis and quality of life in patients with LC. To verify this possibility, large-scale, prospective, randomized and controlled clinical trials should be conducted.

In summary, carnitine deficiency is often found in patients with LC. Carnitine replacement therapy should be administered to LC patients with hepatic encephalopathy, sarcopenia, and muscle cramps, who are deficient in carnitine.

## Figures and Tables

**Figure 1 nutrients-12-01915-f001:**
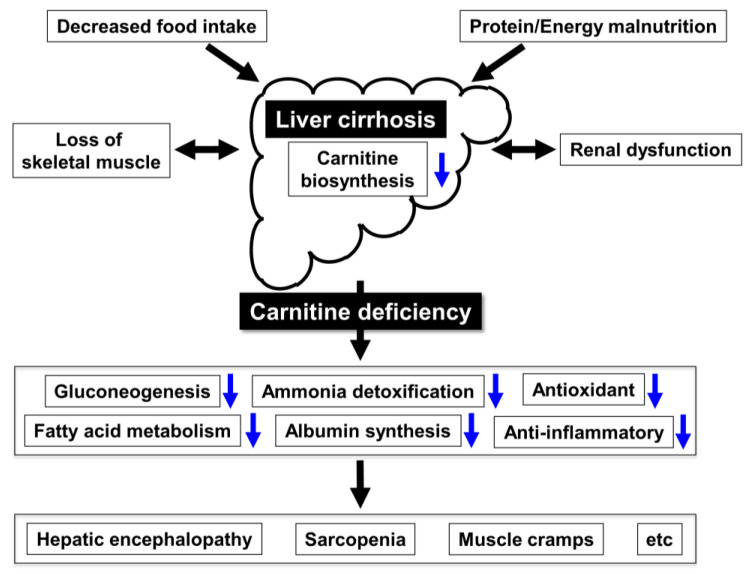
Carnitine deficiency and liver cirrhosis.

**Table 1 nutrients-12-01915-t001:** Physiological function of carnitine.

1. Promotion of energy metabolism and adenosine triphosphate production by β-oxidation of long-chain fatty acids (fatty acid metabolism)
2. Control of various metabolic pathways (gluconeogenesis, urea cycle, glycolysis, tricarboxylic acid cycle, and so on) by maintaining intracellular free coenzyme A (CoA) pool
3. Exclusion of acyl-CoA to the outside of the cell (endogenous detoxification)
4. Antioxidant effect
5. Anti-apoptotic effect
6. Anti-inflammatory effect
7. Biomembrane-stabilizing effect
8. Anti-fibrosis effect and others

**Table 2 nutrients-12-01915-t002:** Carnitine concentration in human organs and tissues.

Blood/Organ	Carnitine Concentrations	Carnitine Concentrations
		(Cirrhosis)
Plasma Total Carnitine	40–60 μmol/L	68.4 ± 4.7 μmol/L
Plasma Free Carnitine	35–50 μmol/L	53.2 ± 2.6 μmol/L
Plasma AcylCarnitin	<15 μmol/L	13.2 ± 1.1 μmol/L
Liver	1000–1900 nmol/g tissue	2100 ± 400 nmol/g tissue
Heart	3500–6000 nmol/g tissue	
Skeletal muscle	2000–4600 nmol/g tissue	
Kidney	200–500 nmol/g tissue	

Cited and modified from the papers by Stanley and Shiraki et al. [8,10].

**Table 3 nutrients-12-01915-t003:** Symptoms in cirrhosis patients with carnitine deficiency.

1. Hepatic encephalopathy
2. Sarcopenia
3. Muscle cramps
4. Appetite loss
5. Sleep disturbance
6. Malaise
7. Impaired consciousness
8. Cardiac dysfunction

**Table 4 nutrients-12-01915-t004:** Differing doses of carnitine supplementation in various studies.

Study Author.	Year	Region	Study Type	N	Intervention	Duration	Outcome Measures	Results
Hepatic encephalopathy								
Malaguarnera, et al. [26]	2003	Italy	RCT	120	Carnitine (4000 mg/day) vs. placebo	2 months	Ammonia levels	*p* < 0.01
							NCT-A	*p* < 0.001
Malaguarnera, et al. [27]	2005	Italy	RCT	150	Carnitine (4000 mg/day) vs. placebo	3 months	Ammonia levels	*p* < 0.05
							MHE and HE 1 or 2	*p* < 0.05
Malaguarnera, et al. [28]	2006	Italy	RCT	24	Carnitine (4000 mg/day) vs. placebo	3 days	Ammonia levels	*p* < 0.05
						6 hours	Glasgow Coma Scale	*p* < 0.001
Malaguarnera, et al. [29]	2008	Italy	RCT	125	Carnitine (4000 mg/day) vs. placebo	3 months	Ammonia levels	*p* < 0.001
							MHE	*p* < 0.001
Malaguarnera, et al. [30]	2009	Italy	RCT	48	Carnitine (4000 mg/day) plus BCAA (20g/day)	1 day	Ammonia levels	*p* < 0.01
					vs. BCAA only (40g/day)		Glasgow Coma Scale	*p* < 0.05
Malaguarnera, et al. [31]	2011	Italy	RCT	121	Carnitine (4000 mg/day) vs. placebo	3 months	Ammonia levels	*p* < 0.001
							QOL	*p* < 0.05
							Physical activity	*p* < 0.05
Malaguarnera, et al. [32]	2011	Italy	RCT	61	Carnitine (4000 mg/day) vs. placebo	3 months	Ammonia levels	*p*< 0.001
							Cognitive functions	*p*< 0.05
Shiraki, et al. [10]	2017	Japan	Retrospective	27	Carnitine (1800 mg/day)	3 months	Ammonia levels	*p* < 0.05
Nojiri, et al. [33]	2018	Japan	RCT	76	Carnitine (1200 mg/day) vs. placebo	3 months	MHE	*p* < 0.05
							Ammonia levels	*p* < 0.05
Sarcopenia								
Malaguarnera, et al. [31]	2011	Italy	RCT	121	Carnitine (4000 mg/day) vs. placebo	3 months	SPPB	*p*< 0.001
Ohara, et al. [34]	2018	Japan	Retrospective	70	Carnitine (1018 mg/day)	11 months	Skeletal muscle mass	*p*< 0.01
Hiramatsu, et al. [35]	2019	Japan	Retrospective	52	Carnitine (≥1274 mg/day)	12 months	Skeletal muscle mass	*p*< 0.05
							Ammonia levels	*p* < 0.05
Muscle cramps								
Nakanishi, et al. [36]	2015	Japan	Prospective	42	Carnitine (900 to 1200 mg/day)	2 months	Muscle cramps	*p* < 0.001
Hiraoka, et al. [37]	2019	Japan	Prospective	18	Carnitine (1000 mg/day) plus exercise	6 months	Muscle cramps	*p* < 0.05

HE, hepatic encephalopathy; MHE, minimal HE; NCT-A, number connection test A; QOL, quality of life; RCT, randomized controlled trial; SPPB, short physical performance battery.

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
