# Peer review of "Usefulness of Carnitine Supplementation for the Complications of Liver Cirrhosis"

_nutrients, 2020, doi:10.3390/nu12071915_

Round 1
Reviewer 1 Report
The authors overviewed the carnitine supplementation for patients with liver cirrhosis. Overall, the manuscript seems to be well-written, but I have some comments to be addressed.
Comments:
1) The authors mentioned ‘Given that the clinical symptoms and examination findings of carnitine deficiency are diverse, it is important to measure the serum carnitine level when such pathophysiology is suspected’. However, they also described ‘We have reported that serum carnitine concentrations (total carnitine, free carnitine, and acylcarnitine) were within the normal levels in the majority of Japanese LC patients’. They should clearly explain why ‘it is important to measure the serum carnitine level when such pathophysiology is suspected’.
2) In relation to my comment No.1, carnitine deficiency may be difficult to be diagnosed only with the blood test. An additional table with major symptoms and its frequencies of carnitine deficiency in cirrhotic patients would be beneficial for readers, as it may help come up with the presence of the carnitine deficiency.
3) The authors mainly mentioned the usefulness of carnitine supplementation. It may be also helpful to discuss the appropriate amount of intake, as high dose of the carnitine intake may cause some adverse events (Rebouche CJ. Carnitine. In: Modern Nutrition in Health and Disease, 9th Edition; edited by Shils ME, Olson JA, Shike M, Ross, AC. Lippincott Williams and Wilkins, New York, 1999, pp. 505-12.)
Author Response
Responses to Reviewer 1
Thank you very much for reviewing our manuscript and offering valuable advice. We appreciate your comments, which have helped us to improve our manuscript. Please find below detailed responses to the reviewer’s comments.
- The authors mentioned ‘Given that the clinical symptoms and examination findings of carnitine deficiency are diverse, it is important to measure the serum carnitine level when such pathophysiology is suspected’. However, they also described ‘We have reported that serum carnitine concentrations (total carnitine, free carnitine, and acylcarnitine) were within the normal levels in the majority of Japanese LC patients’. They should clearly explain why ‘it is important to measure the serum carnitine level when such pathophysiology is suspected’.
We realized that the suggested sentence was unclear and may lead to misunderstanding regarding the diagnosis of carnitine deficiency in patients with cirrhosis. Since the blood carnitine level does not always decrease in patients with liver cirrhosis, its diagnosis relies mostly on the clinical symptoms such as hepatic encephalopathy, sarcopenia, muscle cramps, and so on. For these reasons, we deleted the pointed sentence and revised the text (lines 94 to 98). We thank the reviewer’s valuable suggestion.
- In relation to my comment No.1, carnitine deficiency may be difficult to be diagnosed only with the blood test. An additional table with major symptoms and its frequencies of carnitine deficiency in cirrhotic patients would be beneficial for readers, as it may help come up with the presence of the carnitine deficiency.
Thank you for the useful suggestions. Little evidence exists regarding the frequency of carnitine deficiency in patients with LC. Our previous study involving 70 cirrhosis patients showed that the free carnitine concentration was in the normal range of 53.2 ±2.6 μmol/L [Ref. #14], suggesting the difficulty of diagnosing carnitine deficiency based only on blood test results (lines 94 to 98). Based on the reviewer’s comments, we also revised Table 2 and added new Table 3. These new tables clearly show major symptoms and carnitine concentrations in liver cirrhosis patients.
- The authors mainly mentioned the usefulness of carnitine supplementation. It may be also helpful to discuss the appropriate amount of intake, as high dose of the carnitine intake may cause some adverse events (Rebouche CJ. Carnitine. In: Modern Nutrition in Health and Disease, 9th Edition; edited by Shils ME, Olson JA, Shike M, Ross, AC. Lippincott Williams and Wilkins, New York, 1999, pp. 505-12.)
We understood that there is a need to consider the appropriate amount of carnitine supplementation and note the adverse events that have been associated with high-dose carnitine supplementation. As suggested, there is still a controversy concerning the appropriate amount of carnitine intake. Research has suggested that high-dose carnitine supplementation (approximately 3 g/day) may cause several adverse events including nausea, vomiting, abdominal cramps, diarrhea, muscle weakness, seizures, atherosclerosis, and cardiovascular disease. We added the relevant information and references to the manuscript (lines 142 to 147 and new Refs. #21 to 25), in accordance with the reviewer's comments. We thank the reviewer’s valuable suggestion.
In closing, let me thank you once again for your comments which have helped us to improve the quality of our paper.

Reviewer 2 Report
Dear Authors,
The topic is undoubtedly interesting.
Recently, many publications on the use of carnitine supplementation in liver diseases and their complications have been published. In the current manuscript there are not many cited papers on these topics.
Increasing the value of manuscript is always placing a table summarizing the results of many publications. Please also note that different doses of carnitine have been administered in previous studies. It will be beneficial for readers.
Author Response
Responses to Reviewer 2
Thank you very much for reviewing our manuscript and offering valuable advice. We appreciate your comments, which have helped us to improve our manuscript. Please find below detailed responses to the reviewer’s comments.
- Recently, many publications on the use of carnitine supplementation in liver diseases and their complications have been published. In the current manuscript there are not many cited papers on these topics.
In accordance with the reviewer's comments, we revised the text by citing new references (lines 142 to 147 and new Refs. #21 to 25). In the revision, we initially described the adverse events that have been associated with high-dose carnitine supplementation reported by several clinical trials. We also added new Table 4 that summarizes the information on carnitine supplementation dose, duration, and outcomes reported by previous studies (lines 147 to 148 and new Table 4). We believe these new description and table are useful for readers to deeply understand the content of the paper. We thank the reviewer’s valuable suggestion.
- Increasing the value of manuscript is always placing a table summarizing the results of many publications. Please also note that different doses of carnitine have been administered in previous studies. It will be beneficial for readers.
Based on this valuable suggestion, we added new Table 4 that summarizes the information on carnitine supplementation dose, duration, and outcomes reported by previous studies (lines 147 to 148 and new Table 4). We believe that this new table is helpful for readers to understand the content of our manuscript. We greatly appreciate the reviewer’s important comment again.
In closing, let me thank you once again for your comments which have helped us to improve the quality of our paper.

Round 2
Reviewer 1 Report
The paper was revised according to the reviewers' comments.
Reviewer 2 Report
I accept revised manuscript